# Measurement of Indoor-Outdoor Carbonyls in Three Different Universities Located in the Metropolitan Zone of Mexico Valley during the First Period of Confinements Due to COVID-19

**Rocío García** [1,*], **Sandra Silva Gómez** [2], **Gema Andraca** [1], **Ricardo Torres Jardón** [1], **Agustín García Reynoso** [1], **Julia Griselda Cerón** [3], **Rosa María Cerón** [3] and **Violeta Mugica Alvarez** [2]

1   Institute of Atmospheric Sciences ad Climatic Change, National University of Mexico, Mexico City 04510, Mexico
2   Department of Basic Sciences, Metropolitan Autonomous University, Azcapotzalco Campus, Av. San Pablo 180, Mexico City 02200, Mexico
3   Chemistry Faculty, Autonomous University of Carmen, Calle 56 No. 4, Ciudad del Carmen 24180, Mexico
*   Correspondence: gmrocio@atmosfera.unam.mx

**Abstract:** Carbonyl concentrations in indoor-outdoor air were measured at three urban sites in the Metropolitan Zone of Mexico Valley (MZMV) during the first period of confinements due to COVID-19; the exposure for people living in indoor environments was also assessed. Indoor and outdoor samples were simultaneously collected sequentially with Sep-Pack DNPH-Silica cartridges. Formaldehyde, acetaldehyde, acetone, propionaldehyde, butyraldehyde and acrolein were quantified according to the US-EPA TO-11A method. Acrolein and acetone were the most abundant carbonyls in indoor air, with average concentrations of 55.5 $\mu g\,m^{-3}$ and 46.4 $\mu g\,m^{-3}$, respectively, followed by formaldehyde (29.1 $\mu g\,m^{-3}$), acetaldehyde (21.4 $\mu g\,m^{-3}$) and butyraldehyde (7.31 $\mu g\,m^{-3}$). Propionaldehyde was not detected. Acetone was the dominant carbonyl in outdoor samples with an average concentration of 8.4 $\mu g\,m^{-3}$, followed by formaldehyde (2.8 $\mu g\,m^{-3}$) and acetaldehyde (0.7 $\mu g\,m^{-3}$). Butyraldehyde and acrolein were not detected in outdoor air. Indoor/outdoor (I/O) ratios showed that indoor sources prevail for most aldehydes. Statistical analysis of simple correlations showed that the measured carbonyls were influenced by the presence of indoor sources. The average cancer risk coefficients (LTCR) for formaldehyde and acetaldehyde and the non-cancer risk coefficients (HQ) for acrolein and formaldehyde were higher than the recommended limits, which should be a cause for public concern.

**Keywords:** carbonyl compounds; indoor-outdoor concentrations; emission sources; COVID-19

## 1. Introduction

Aldehydes are a product of the oxidation of almost all hydrocarbons present in the air, these come from anthropogenic activities (burning of fossil fuels, industrial activities, waste, etc.), biomass burning (agricultural burning or forest fires) or natural emissions derived from the metabolism of vegetation (mainly isoprene). They are emitted directly into the atmosphere by vehicles with internal combustion engines [1,2], by gasoline storage tanks, incinerators, boilers and petrochemical processes. They are also formed indirectly in atmospheric reactions being products of the photo-oxidation of hydrocarbons in gas phase [3–5]. Aldehydes are precursors of ozone ($O_3$), volatile organic compounds (VOCs) and nitrogen oxides ($NO_x$) [6–9]. According to some studies [10–13], biogenic sources probably increase secondary formaldehyde and acetaldehyde, and forest fires probably increase secondary acrolein. On the other hand, the increased use of fuel additives such as methyl tert-butyl ether (MTBE) and methyl tert-amyl ether (TAME), or fuels such as methanol (since it is considered a low cost alternative fuel), have increased C1-C4 aldehydes and volatile organic compounds emissions in ambient air [14–16]. LP gas (liquefied propane

70%) is a source of aldehydes because butanes and olefins (e.g., ethylene, propylene and butylene) are more photo-reactive than propane. In this work, pollution in indoor and outdoor environments is associated, considering that the problem of pollution in indoor environments is caused by a variety of emission sources [17–21]. Likewise, ventilation, temperature and humidity are critical factors in determining the concentration levels of aldehydes. Effects of acute or chronic toxicity to formaldehyde and acetaldehyde have been documented, the symptoms range from mild effects in the upper respiratory tract [22], such as nasal congestion, sneezing, acute respiratory diseases, effects such as conjunctivitis, and systemic effects, such as headache [23]. In addition, a substantial portion of outdoor air pollution migrates indoors, affecting its quality.

During the pandemic period, when people had to stay at home for a long time-period, it has been reported that the emission of aldehydes increased in indoor ambient because of an intensive use of antibacterial and antiviral liquids (alcohol base) exacerbating the effects associated with COVID-19 [24]. However, in indoor ambient, besides carbonyl sources, it is necessary to consider that there are some building and construction materials and surfaces that can have a benefic effect as antibacterial or antiviral agents. Such is the case of graphene, fullerene, carbon dots and graphitized oxide sheets, that have shown remarkable antiviral properties as inhibitors of viral activity, or as photoactivators [25,26]. Therefore, materials present in indoor ambient could play a paradoxical role in the effects derived from both carbonyls and secondary organic aerosols sources, and COVID-19, during the pandemic, acting in synergy or exacerbating the health effects observed but also acting as antibacterial or antiviral surfaces. However, more studies about this topic are required to be able to draw a conclusion on this.

During the COVID-19 pandemic, some studies reported aldehyde levels in indoor ambient almost three times greater than those found before the pandemic [27]. This is because of the increase of intensive cleaning activities and the use of hydroalcoholic gels as anti-COVID-19 measures, since aldehydes are known as degradation products of primary alcohols. In addition, air pollution and meteorological factors influence the trends of respiratory disease outbreaks by altering host immunity and pathogen survival time. Since the major route of SARS-CoV-2 transmission is through respiratory droplets of the infected people, there is a plausible association of ambient air pollutants such as particles and aerosols in the viral transmission and related mortality. As a cofactor, atmospheric aerosols can induce indirect systemic effects in the human body and are associated with pro-inflammatory and oxidative mechanisms in the lungs, as well as altered immune system pathways. In addition, it has been suggested that particulate matter and aerosols serve as a carrier for droplet nuclei, increasing the spread of SARS-CoV-2 [24]. In conditions of atmospheric stability and high concentrations of particulate matter and aerosols, there is a high probability of viruses creating clusters with the particles by reducing their diffusion coefficient, enhancing their permanence time and amount in the atmosphere, and promoting contagion [28].

In spite of the major pathways for the degradation of carbonyls in the atmosphere being well known (chemical reactions in the gas phase and photolysis, due to their relatively low Henry's constants, which suggests their water-insoluble properties) [29], other removal pathways, such as heterogeneous reactions in the presence of atmospheric water, could be important for dicarbonyls as glyoxal and methylglyoxal [30–32] play a significant role in the formation of secondary organic aerosols. In addition, it is interesting to discuss the role that humidity plays in the atmosphere, not only in the exacerbation of the possible health effects associated, but also in the transmission process of COVID-19. The importance of this last topic can be explained by two ways. First, since aldehydes contribute to the formation of inorganic aerosols and secondary organic aerosols through phase gas and heterogeneous reactions, it has been established that aerosols and particulate matter could then act as a potential "carrier" for droplet nuclei, triggering a boost effect on the spread of the virus [24]. Second, with regard to climatic factors, it is well known that coalescence phenomena require optimal conditions of temperature and humidity to stabilize the aerosols in the

air; studies have shown that, for coronaviruses and influenza viruses, survival is generally higher at low temperature and low values of absolute humidity [33,34]. Thus, the decrease of COVID-19 growth rate in warm and very humid climates can be explained by the fact that coronavirus persistence outside the organisms decreases at high temperature, medium humidity, and under sunlight; at the same time, the host susceptibility can be higher in cold and dry environments because of a slower mucociliary clearance, or a decreased host immune function.

There are not enough studies that associate indoor and outdoor aldehydes concentrations in Mexico. In the last three years, due to home confinement associated to the COVID-19 pandemic, there was a growing concern about indoor air quality in work places, homes, and public spaces, and the possible exacerbation of respiratory diseases associated to COVID-19 by indoor pollutants. The present work consisted of measuring the levels of carbonyls both in indoor and outdoor air in three different universities in Mexico City during the first period of confinement due to the COVID-19 pandemic, and determining the exposure and risk that these compounds represent in the exposed population.

## 2. Materials and Methods

### 2.1. Sampling Sites

The evaluation of the sources and processes that affect indoor-outdoor air quality requires the selection of different study areas; for this reason, three urban sites were selected, which are located within the facilities of three different universities in the Metropolitan Zone of Mexico Valley (MZMV). The main differences between the three sites involve the following aspects: (a) natural emissions occurrence; and (b) anthropogenic emissions occurrence, such as proximity to industrial areas, vehicular density and influence of anthropogenic sources. Taking into consideration the influence of the factors mentioned above, some of the characteristics of each sampling site are described below. The University-1 site (UAM), is located at $19°30'11''$ north latitude and $99°11'13''$ west longitude, it is a two-story building surrounded by avenues of intense vehicular traffic, it is an area of great industrial and commercial activity, surrounded by important housing complexes located in the northern zone of the MZMV. The University-2 site (UDLA), is located at $19°25'20''$ north latitude and $99°09'59''$ west longitude, it is a three-story building, it is located in an area of great commercial and tourist activity, surrounded by avenues of intense vehicular traffic, located in the downtown area of the MZMV. University-3 site (UNAM), is located at $19°20'01''$ north latitude and $99°11'54''$ west longitude, it is a three-story building, it is located in an area of large green areas, surrounded by research buildings and avenues with intense vehicular traffic, located in the southern zone of the MZMV (Figure 1 and Table 1).

### 2.2. Sampling Method

The sampling was carried out in three seasonal periods, spring, summer and fall; sampling was carried out simultaneously in indoor and outdoor environments. The samples were collected sequentially for approximately 8 h, at intervals of 2 h at a controlled flow rate of 1 L min$^{-1}$ with Sep-Pack DNPH-Silica cartridges (Waters Corp., Milford, MA, USA). A totoal of 246 samples were obtained, the sampling was carried out during the hours of greatest vehicular traffic and industrial and labor activity in the MZMV, from 8:00 to 16:00 h at the three sampling places.

Formaldehyde, acetaldehyde, acetone, propionaldehyde, butyraldehyde and acrolein were quantified according to the certified method TO-11A (US-EPA) [35]. In addition, in order to eliminate the interference caused by ozone ($O_3$), a copper coil was placed internally impregnated with potassium iodide solution (KI) at 10% avoiding the degradation of hydrazone derivatives, during the determinations of aldehydes [36,37]. Collection efficiency was determined by connecting two cartridges in a series under the same sampling conditions described, obtaining values > 95% for all the carbonyls analyzed. The determination is based on the reaction of these compounds in a sensitive absorbing solution of 2,4-DNPH.

After the completion of the reaction, the derivatives were analyzed by the HPLC system without any post sample preparation [38–42].

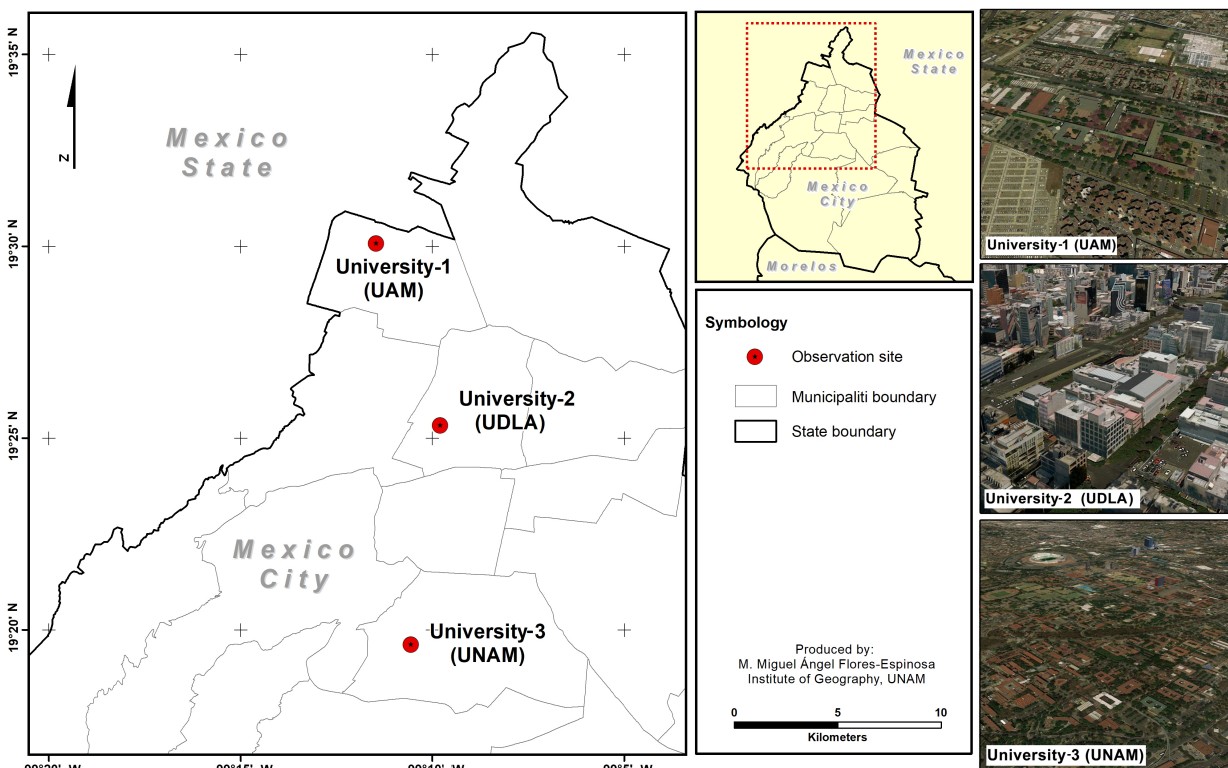

**Figure 1.** Location and geographical characteristics of the study sites.

**Table 1.** Characteristics of the study sites: University-1 (UAM), University-2 (UDLA) and University-3 (UNAM).

| Site | Ventilation | Characteristics |
|---|---|---|
| University-1 (UAM) | Through doors and windows. | Two-story building, walls painted with vinyl paint, granite floors, wood chipboard furniture. No smoking building. |
| University-2 (UDLA) | Through doors and windows. | Three-story building, walls painted with vinyl paint, granite floors, wood chipboard furniture. On the roof of the building there is an exclusive area for smoking. |
| University-3 (UNAM) | Through doors and windows. | Three-story building, walls painted with vinyl paint, tile floors, wood-paneled cabinets. No smoking building. |

### 2.3. Quality Assurance

Field and laboratory blanks were evaluated. To ensure the quality of the analyses, detection limits and reproducibility were determined. Method detection limits (MDLs) were defined as three times the standard deviation of the blanks. Detection limits for formaldehyde, acetaldehyde, acetone, propionaldehyde, butyraldehyde and acrolein derivatives were $1 \times 10^{-4}$, $1 \times 10^{-4}$, $2 \times 10^{-4}$, $1 \times 10^{-4}$, $1 \times 10^{-4}$ and $3 \times 10^{-4}$ ppm, respectively. Calibration curves ranging from 0.05 to 0.405 ppm were constructed with a mixture of formaldehyde, acetaldehyde, acetone, propionaldehyde, butyraldehyde and acrolein standards, diluted with acetonitrile of 15 µg mL$^{-1}$ Supelco CRM47285 standard solution. Finally, from the concentration of each carbonyl compound in ppm and the volume of air sampled, which in this case was 1 L min$^{-1}$, the concentration in ppbv of each compound was calculated.

### 2.4. Analytical Method

The sampled cartridges were taken to the laboratory and diluted with a pure solution of acetontrile favoring the derivatization reaction to form 2,4-dinitrophenylhydrazones, commonly referred to as hydrazones. The analysis was performed in a Shimadzu model SPD-20 AV High Resolution Liquid Chromatograph (HPLC) with a UV-visible detector at a wavelength of 360 nm with a column in reversed phase, ACE5 C18, 5 μm, 25 cm × 4.6 mm I.D.

### 2.5. Statistic Analysis

A Friedman test was applied to the data set in order to investigate if there were significant differences in carbonyls concentrations among the sampling sites and sampling periods. In addition, a bi-variate analysis was carried out in order to obtain Spearman's correlation coefficients and relations between pairs of carbonyls. All the statistical analysis was carried out using XLSTAT (Basic+ Version by Addinsoft Inc, New York, USA) and SPSS (Faculty Pack by IBM Inc, New York, USA) tools.

### 2.6. Exposure ad Risk Assessment

Formaldehyde has been classified by IARC as a carcinogen within group 1: carcinogenic to humans [43] based on limited evidence of carcinogenicity in humans and sufficient evidence of carcinogenicity in animals. Acetaldehyde has been classified within group 2B (probably carcinogenic to humans) due to inadequate evidence for human carcinogenicity and sufficient evidence for animal carcinogenicity, and has been linked to the development of tumors in the respiratory tract in animals [44]. Acrolein has been classified within group 2A (possibly carcinogenic to humans) [45]. To assess the risk to which the inhabitants of the study areas may be exposed by inhalation, the non-cancer risk coefficient, and the lifetime cancer risk coefficient (LTCR) due to exposure to formaldehyde, acetaldehyde and acrolein at the found levels in this study, were estimated according to the methodology described by Zhang et al. [46]. Daily exposure (*E*) in milligrams per kilogram per day of an individual by inhalation can be calculated as follows:

$$E = C \times I_{RA} \times \frac{D_a}{B_{wa}}, \tag{1}$$

where $C$ (mg m$^{-3}$) is the carbonyl compound concentration, $I_{RA}$ is the inhalation rate for an adult (0.83 m$^3$ h$^{-1}$) according to USEPA-IRIS [47], $D_a$ is the exposure duration for an adult (24 h day$^{-1}$ for indoor air and 16 h day$^{-1}$ for outdoor environments) and $B_{wa}$ is the body weight for an adult (65 kg) [48].

The lifetime cancer risk coefficient (*LTCR*) is calculated by the following equation:

$$LTCR = E \times SF, \tag{2}$$

where *SF* is the slope factor of inhalation unit risk for air toxics when the exposure-carcinogenic effect is considered linear. *SF* values for measured carbonyls are showed in Table 2 and taken from US-EPA [49]. *LTCR* values should not exceed the recommended limits established by EPA ($1 \times 10^{-6}$) and WHO ($1 \times 10^{-5}$). Otherwise, the exposed population is at risk of developing cancer in their lifetime.

The non-cancer risk coefficients (risk of developing cardiovascular and respiratory diseases) were determined as hazard quotients (*HQ*):

$$HQ = \frac{C}{Rfc}, \tag{3}$$

where $C$ is the average daily-received concentration and *Rfc* is the inhalation reference concentration for each carbonyl compound (Table 2) which were taken from US-EPA [50,51]. According to WHO and US-EPA, if $HQ > 1.0$, it indicates that long-term exposure may result in adverse health effects (cardiovascular and respiratory diseases).

**Table 2.** Toxicological profiles and parameters for carbonyl compounds.

| Carbonyl Compounds | CAS No. | Reference Concentration RfC [1] (mg/m$^3$) | Inhalation Cancer Slope Factor SF [2] | Cancer Classification [3] |
|---|---|---|---|---|
| Formaldehyde | 50-00-0 | $9.83 \times 10^{-3}$ | $2.1 \times 10^{-2}$ | Group 1 |
| Acetaldehyde | 75-07-0 | $9.00 \times 10^{-3}$ | $1 \times 10^{-2}$ | Group 2 B |
| Acrolein | 107-02-8 | $2 \times 10^{-5}$ | - | Group 2 A |
| Acetone | 67-64-1 | - | * | ** |
| Propionaldehyde | 123-38-6 | $8 \times 10^{-3}$ | * | ** |
| Butyraldehyde | 123-72-8 | - | * | ** |

[1] Integrated Risk Information System (EPA-IRIS) [50,51]; [2] The Risk Assessment Infotrmation System (RAIS) [49]; [3] International Agent for Research on Cancer (IARC) [42,43]; Information reviewed but value not estimated within EPA-IRIS Program; * Not assessed under the IRIS Program; ** Not classified by IARC.

## 3. Results and Discussion

### 3.1. Diurnal and Seasonal Variation of Carbonyls

The results showed that acrolein and acetone were the most abundant carbonyl compound in indoor air, with an average concentration of 55.5 µg m$^{-3}$ and 46.4 µg m$^{-3}$, respectively, followed by formaldehyde (29.1 µg m$^{-3}$) and acetaldehyde (21.4 µg m$^{-3}$) and butyraldehyde (7.31 µg m$^{-3}$). Propionaldehyde was not detected in indoor air samples. Acetone was most abundant in all sampling places in outdoor air, with an average concentration of (8.4 µg m$^{-3}$), followed by formaldehyde (2.8 µg m$^{-3}$) and acetaldehyde (0.7 µg m$^{-3}$). Butyraldehyde and acrolein were not detected in outdoor air samples, acetone was the most abundant carbonyl in this study as is expected since this compound dominates the composition of oxygenated volatile organic compounds in the atmosphere [52,53] and has a relatively long residence time (15–35 days), so its permanence in the atmosphere is large [53]. In addition, acetone has several sources that include direct emissions, oxidation of anthropogenic hydrocarbons, mainly from propane, isobutene and isobutane (this source provides the single largest source of acetone in the atmosphere: 51%). In this regard, in the 2018 emissions inventory of MZMV, large emissions of volatile organic compounds were reported derived from leaks at liquefied petroleum gas facilities (LP gas) (32,824.24 t year$^{-1}$) and storage and distribution facilities of this kind of gas (1,646.51 t year$^{-1}$) [54]. It is known that LP gas distributed in Mexico is a mixture of propane, butane, isobutane, ethane and methane, so the oxidation mainly of the C$_3$ and C$_4$ hydrocarbons contained in this gas may be the main cause of the high concentrations of acetone measured in the study area. Table 3 shows the total carbonyls concentration of indoor-outdoor environments in the three sampling sites: University-3 (UNAM), University-2 (UDLA) and University-1 (UAM).

**Table 3.** Total carbonyls concentration of indoor-outdoor environments: University-3 (UNAM), University-2 (UDLA) and University-1 (UAM).

| Concentration (µg m$^{-3}$) | University-1 (UAM) | University-2 (UDLA) | University-3 (UNAM) |
|---|---|---|---|
| Total carbonyls Indoor | 187.6 | 531.7 | 836 |
| Total carbonyls Outdoor | 45 | 63.8 | 89 |

#### 3.1.1. Indoor-Outdoor Carbonyls Concentrations

The results of the Friedman tests show that there were significant differences between the concentration values of all carbonyl compounds, except for acrolein, between the three sampling sites. This confirmed that the nature of its sources is different between each site. Additionally, it was observed that there were significant differences in the concentrations of acrolein, acetaldehyde, formaldehyde and propionaldehyde among the different sampling campaigns (spring and summer with fall). Formaldehyde, acrolein, acetone, propionaldehyde and butyraldehyde (excepting during fall 2019) had higher mean con-

centrations in indoor environments (Table 4). It is well known that average concentrations of total carbonyl observed outdoors are much lower than those observed indoors due to the limited dispersion that prevails in indoor ambient. Acetaldehyde showed higher mean concentrations values in outdoor environments, excepting during spring 2019. In indoor environments, the mean values of concentration were higher for formaldehyde acetaldehyde and acetone during the summer sampling campaign. Acrolein and propionaldehyde showed higher mean concentrations during the fall sampling campaign, whereas butyraldehyde showed higher mean values during the spring sampling campaign for indoor environments. For outdoor environments, acetaldehyde, acrolein and acetone showed higher mean concentrations during the summer campaign, whereas formaldehyde, propionaldehyde and butyraldehyde had higher mean values during the fall campaign.

**Table 4.** Mean values of indoor-outdoor carbonyl concentrations during the three different sampling campaigns.

| Campaign | Carbonyls Concentrations ($\mu g/m^3$) | | | | | | | | | | | |
|---|---|---|---|---|---|---|---|---|---|---|---|---|
| | FA Indoor | FA Outdoor | AA Indoor | AA Outdoor | ACR Indoor | ACR Outdoor | AC Indoor | AC Outdoor | PR Indoor | PR Outdoor | BUT Indoor | BUT Outdoor |
| Spring 2019 | 2.53 | 0.94 | 1.95 | 1.34 | 0.46 | 0.37 | 68.58 | 4.18 | 0.42 | 0.33 | 5.18 | 0.95 |
| Summer 2019 | 5.59 | 1.32 | 7.41 | 7.80 | 2.46 | 4.49 | 97.23 | 7.24 | 0.72 | 0.40 | 1.84 | 1.44 |
| Fall 2019 | 4.41 | 2.39 | 2.84 | 4.90 | 8.47 | 0.63 | 27.84 | 9.71 | 1.27 | 1.15 | 1.79 | 1.94 |

FA: Formaldehyde; AA: Acetaldehyde; ACR: Acrolein; AC: Acetone; PR: Propionaldehyde; BUT: Butyraldehyde.

Figures 2–4 show the seasonal variation of the contribution percentage of each carbonyl compound to the total carbonyl concentrations in University-1 (UAM), University-2 (UDLA) and University-3 (UNAM), considering both indoor and outdoor concentrations.

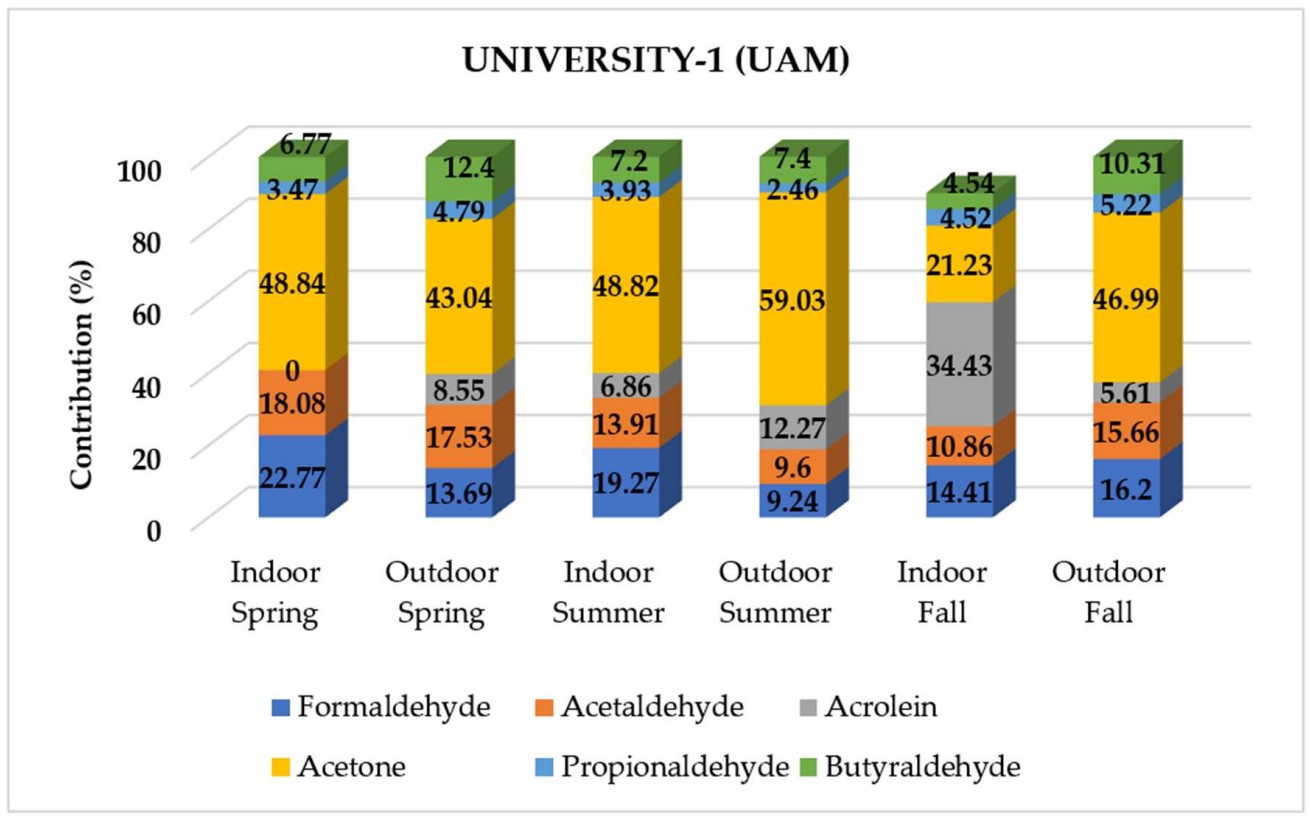

**Figure 2.** Seasonal variation of the contribution percentage of each carbonyl compound to the total carbonyl concentrations (indoor and outdoor) in University-1 (UAM).

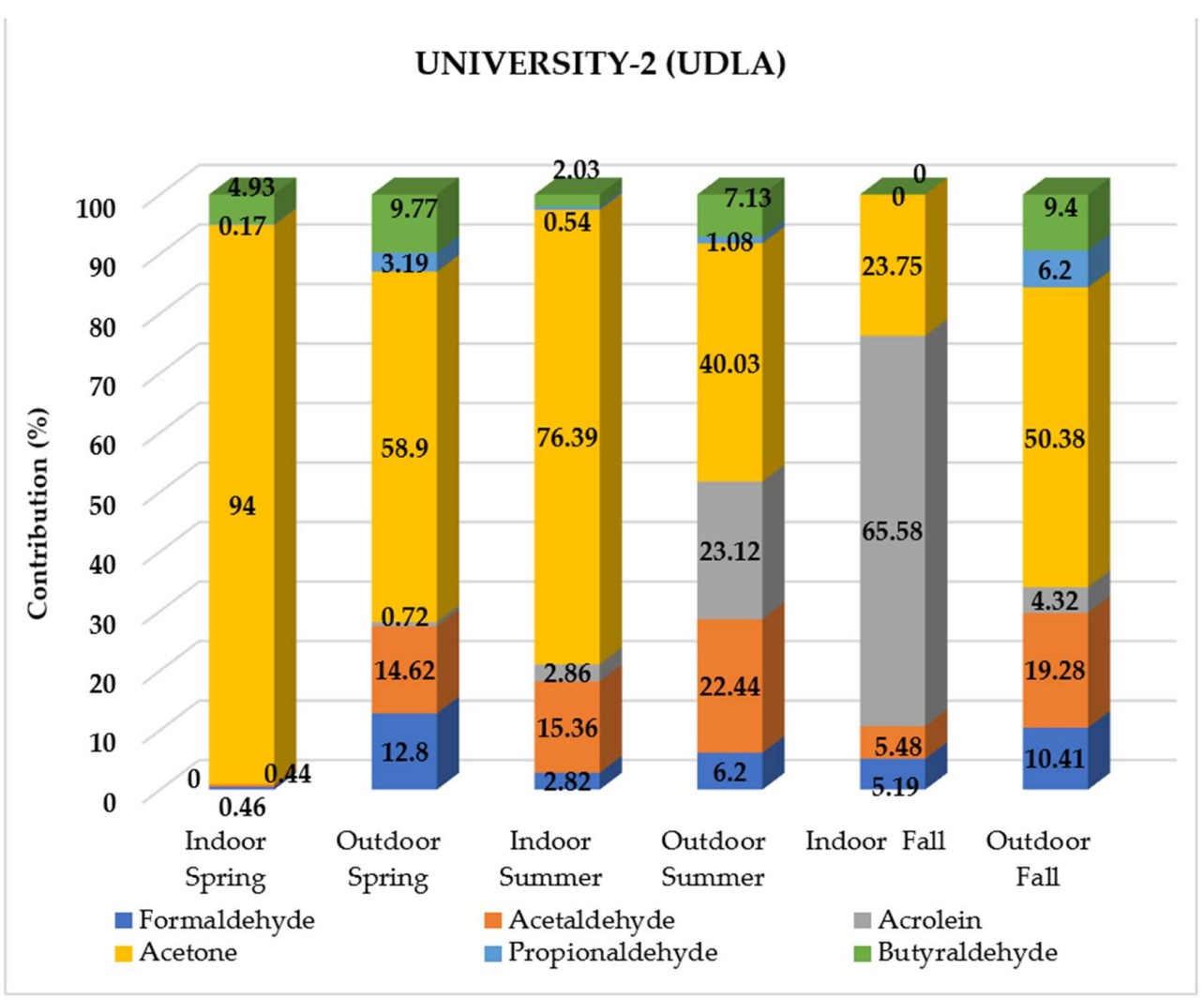

**Figure 3.** Seasonal variation of the contribution percentage of each carbonyl compound to the total carbonyl concentrations (indoor and outdoor) in University-2 (UDLA).

University-1 (UAM)

Acetone contributed 48.84% and 43% to the total carbonyls during the spring campaign, in indoor and outdoor environments, respectively, followed by acetaldehyde, formaldehyde and butyraldehyde (Figure 2). During the summer, acetone contributed with 48.82% and 59.08% to the total carbonyls for indoor and outdoor environments, respectively. There was an important contribution of formaldehyde (19.27%) in indoor environments and acrolein (12.27%) in outdoor environments (Figure 2). During the fall sampling period, in indoor environments, acrolein and acetone contributed 34.43% and 21.23% to the total carbonyls, whereas in outdoor environments, acetone and formaldehyde were the main contributors to the total carbonyls with 46.99% and 16.2%, respectively (Figure 2).

University-2 (UDLA)

In the University-2 sampling site, during spring (Figure 3), the main contributors to the total carbonyls were acetone (94%) in indoor environments and acetone (58.9%) and acetaldehyde (14.62%) in outdoor environments. During summer (Figure 3), acetone and acetaldehyde contributed 76.39% and 15.36%, respectively, in indoor environments, whereas in outdoor environments, the main contributors to the total carbonyls were acetone and acrolein with 40.03% and 23.12%, respectively. During the fall sampling period (Figure 3), acrolein and acetone were the main contributors to the total carbonyls in indoor

environments with 65.58% and 23.75%, respectively. In outdoor environments during the fall campaign, the main contributors to the total carbonyls were acetone (50.38%) and acetaldehyde (19.28%) for this sampling site.

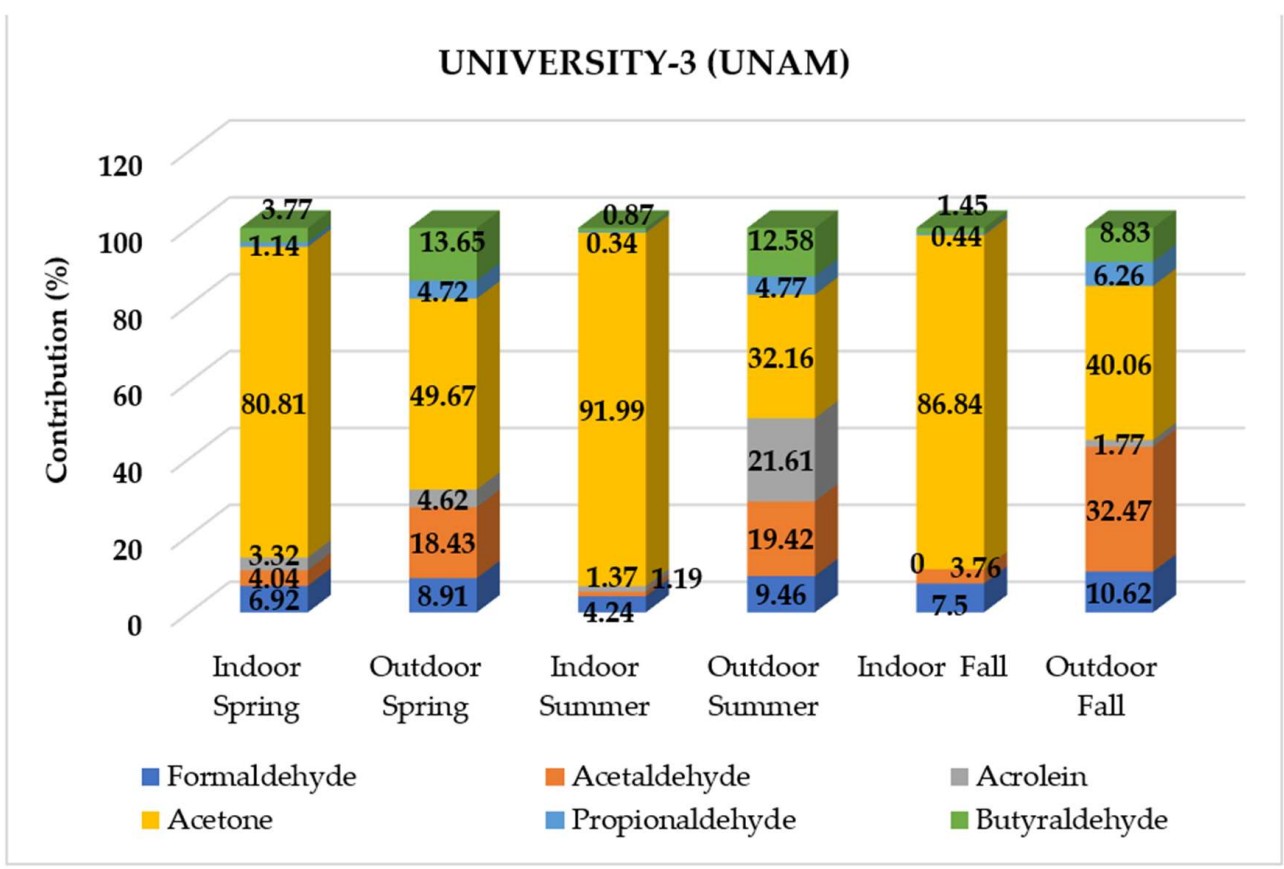

**Figure 4.** Seasonal variation of the contribution percentage of each carbonyl compound to the total carbonyl concentrations (indoor and outdoor) in University-3 (UNAM).

University-3 (UNAM)

The main contributors to the total carbonyls in this site were formaldehyde, acetaldehyde and acetone. During the spring (Figure 4), acetone and formaldehyde contributed 80.81% and 6.92% to the total carbonyls in indoor environments, whereas in outdoor environments, acetone and acetaldehyde were the main contributors with 49.67% and 18.43%, respectively. During the summer campaign (Figure 4), acetone was the main contributor to the total carbonyls in indoor environments in this site (91.99%), whereas in outdoor environments, the main contributors were acetone (31.1%) and acetaldehyde (19.42%). During the fall sampling period (Figure 4), acetone and formaldehyde contributed to the total carbonyls in indoor environments with 86.84% and 7.5%, respectively. In outdoor environments, the main contributors to the total carbonyls during the fall campaign were acetone (40.06%) and acetaldehyde (32.47%).

3.1.2. Diurnal and Seasonal Variation of Indoor Carbonyl Concentrations

Table 5 shows the diurnal and seasonal variation of indoor carbonyl concentrations for the three sampling sites. In studies carried out in residences, formaldehyde is usually reported as the most abundant carbonyl compound, followed by acetaldehyde, acetone and acrolein [55]. However, in this study, the dominant carbonyl in indoor environments was acetone. Acetone showed the highest indoor concentrations during the spring and summer sampling campaigns in University-2 (UDLA) and University-3 (UNAM). It is well known that average concentrations of total carbonyl observed outdoors are much lower than those

observed indoors due to the limited dispersion that prevails in indoor ambient. In addition, since during the pandemic period people had to stay at home for long time-periods, people were more exposed to these toxic compounds. For this reason, some studies have reported carbonyl concentrations in indoor ambient during the COVID-19 pandemic higher than those found before the pandemic. For example, Ninya et al. (2022) [27] carried out a study in different places in Tarragona, Spain. They compared carbonyl levels indoors in the same sampling sites during 2021 (during the COVID-19 pandemic) with those obtained during 2019 (before the pandemic), finding that concentrations indoors were three times higher during the pandemic period. This is because of the more intensive cleaning activities and the frequent use of alcohol-based sanitizers and hydro alcoholic gels as anti-COVID-19 measures. It is well known that aldehydes and ketones are degradation products of primary alcohols, therefore, it is expected that during the pandemic period, in which these compounds were intensively used, carbonyl concentrations had increased. This could explain why, in our study, carbonyl concentrations, specifically in the case of acetone, were higher indoors and also higher during spring and summer campaigns, further considering that during these warm seasons, solvent evaporation and emissions from surfaces and household materials could be higher too. On the other hand, it can be observed that acetone concentrations were higher in the University-2 site; regarding this, it is necessary to consider that in spite of anti-COVID-19 protocols being applied in all public buildings and schools, the frequency of application of sanitization protocols was closely related to the economic resources available. In the case of University-2 (UDLA), it is a small and private school where sanitization protocols were more frequent and rigorous, in comparison with UNAM and UAM, which are largest and public schools, in comparison with UDLA. This fact could explain why higher levels of acetone were found in University-2 (UDLA). On the other hand, indoor carbonyls, mainly formaldehyde, is attributed to the age of the buildings, coatings, enamels and glues used in furniture, decorations and carpets. In addition, factors such as temperature and humidity have been associated with the release of carbonyls, due to the effect that they have on furniture surfaces. In the evaluation of carbonyls in residences, it must be taken into account that the indoor combustion processes that lead to the emission of carbonyls are mainly activities in kitchens and tobacco burning [33,34].

University-1 (UAM)

The concentrations obtained in indoor environments for University-1 (UAM) during the spring showed an hourly variation, with a maximum concentration in a period from 10:00 to 14:00 h (Table 5) being acetone the carbonyl, with higher indoor concentrations in this sampling site. During the summer, the peak of indoor carbonyls concentrations was observed from 10 to 12 h, with the highest indoor concentrations for acetone and formaldehyde. For the fall sampling campaign, this site did not show a clear diurnal pattern, with acrolein being the carbonyl with the highest indoor concentration value, followed by acetone, in the sampling period comprised from 10:00 to 12:00 h. Carbonyl indoor concentrations in this site had a clear seasonal pattern, with maximum values during the fall sampling period and registering the lowest values during the spring period.

University-2 (UDLA)

During the spring period, formaldehyde, acetaldehyde, acetone and butyraldehyde had higher indoor concentrations from 12:00 to 14:00 h (Table 5), whereas, acrolein and propionaldehyde showed higher indoor concentrations in the period comprised from 10:00 to 12:00 h. During the summer, all measured carbonyls showed higher indoor concentrations during the period comprised from 10:00 to 12:00 h, except acetone, which had its maximum concentration from 12:00 to 14:00 h (Table 5). Indoor carbonyl concentrations in this site during the fall showed their maximum values in the period comprised from 10:00 to 12:00 h, except acrolein, which had its maximum concentration values from 08:00 to 10:00 h (Table 5). Formaldehyde, acetaldehyde and acrolein showed higher concentrations during the summer campaign, whereas acrolein and propionaldehyde had their maximum

indoor concentration values during the spring period. The lowest indoor concentrations for all carbonyls were registered for this site during the fall sampling period.

**Table 5.** Diurnal variation of carbonyl indoor concentrations at the three sampling sites (University-1 UAM, University-2 UDLA and University-3 UNAM) during the three sampling campaigns.

| University-1 (UAM) | | | | | | | | | | | | | | | | | |
|---|---|---|---|---|---|---|---|---|---|---|---|---|---|---|---|---|---|
| Season: | Spring | | | | | | Summer | | | | | | Fall | | | | | |
| Sampling Period (h) | Indoor Carbonyls Concentrations (µg/m³) | | | | | | | | | | | | | | | | | |
| | FA | AA | ACR | AC | PR | BUT | FA | AA | ACR | AC | PR | BUT | FA | AA | ACR | AC | PR | BUT |
| 8–10 | 3.87 | 2.55 | 0.06 | 5.24 | 0.37 | 1.04 | 3.77 | 2.38 | 1.36 | 6.65 | 0.50 | 2.04 | 7.47 | 5.66 | 1.06 | 17.72 | 1.07 | 2.90 |
| 10–12 | 4.27 | 3.96 | - | 6.03 | 0.63 | 1.15 | 4.75 | 3.64 | 1.71 | 15.30 | 1.09 | 1.61 | 6.64 | 5.74 | 47.36 | 13.59 | 1.65 | 2.32 |
| 12–14 | 3.81 | 3.48 | - | 7.63 | 0.67 | 1.24 | 4.74 | 3.54 | 1.65 | 11.60 | 1.10 | 1.30 | 8.01 | 5.27 | 0.23 | 17.00 | 3.91 | 1.89 |
| 14–16 | 4.05 | 3.18 | - | 6.77 | 0.83 | 1.30 | * | * | * | * | * | * | * | * | * | * | * | * |
| University-2 (UDLA) | | | | | | | | | | | | | | | | | |
| Season: | Spring | | | | | | Summer | | | | | | Fall | | | | | |
| Sampling period (h) | Indoor Carbonyls concentrations (µg/m³) | | | | | | | | | | | | | | | | | |
| | FA | AA | ACR | AC | PR | BUT | FA | AA | ACR | AC | PR | BUT | FA | AA | ACR | AC | PR | BUT |
| 8–10 | 0.80 | 0.89 | 0.28 | 39.76 | 0.20 | 4.67 | 1.20 | 34.23 | 1.52 | 16.05 | 0.08 | 1.28 | 0.02 | 0.04 | 1.97 | 0.15 | - | 1.59 |
| 10–12 | 1.26 | 1.28 | 0.28 | 204.55 | 0.27 | 11.98 | 3.76 | 9.25 | 3.81 | 91.80 | 0.69 | 2.37 | 0.07 | 0.06 | - | 0.31 | - | 2.77 |
| 12–14 | 1.40 | 1.32 | 0.11 | 348.71 | 0.26 | 18.43 | 2.73 | 5.35 | 2.56 | 96.37 | 0.65 | 1.98 | 0.04 | 0.05 | 0.23 | 0.19 | - | 1.25 |
| 14–16 | 1.38 | 1.21 | 0.11 | 159.80 | 0.26 | 17.22 | * | * | * | * | * | * | * | * | * | * | * | * |
| University-3 (UNAM) | | | | | | | | | | | | | | | | | |
| Season: | Spring | | | | | | Summer | | | | | | Fall | | | | | |
| Sampling period (h) | Indoor Carbonyls concentrations (µg/m³) | | | | | | | | | | | | | | | | | |
| | FA | AA | ACR | AC | PR | BUT | FA | AA | ACR | AC | PR | BUT | FA | AA | ACR | AC | PR | BUT |
| 8–10 | 2.12 | 1.33 | 1.88 | 8.48 | 0.37 | 1.26 | 9.74 | 2.54 | 3.35 | 149.23 | 0.58 | 2.38 | 4.75 | 2.42 | - | 51.44 | 0.13 | 0.70 |
| 10–12 | 2.51 | 1.58 | 1.48 | 12.57 | 0.50 | 1.59 | 9.49 | 2.90 | 3.87 | 287.62 | 0.94 | 1.75 | 5.65 | 3.20 | - | 65.92 | 0.34 | 1.13 |
| 12–14 | 2.39 | 1.36 | 0.51 | 7.74 | 0.39 | 1.17 | 10.13 | 2.85 | 2.27 | 200.49 | 0.86 | 1.87 | 7.02 | 3.10 | - | 84.27 | 0.55 | 1.55 |
| 14–16 | 2.46 | 1.25 | 0.68 | 15.70 | 0.30 | 1.12 | * | * | * | * | * | * | * | * | * | * | * | * |

FA: Formaldehyde; AA: Acetaldehyde; ACR: Acrolein; AC: Acetone; PR: Propionaldehyde; BUT: Butyraldehyde; -: Lower than the limit of detection; *: Not measured.

University-3 (UNAM)

Indoor carbonyl concentrations in this site showed a clear seasonal pattern with higher values during summer for formaldehyde, acrolein, propionaldehyde and butyraldehyde, and higher concentrations in indoor environments for acetaldehyde during the fall sampling period. The lowest concentration values for formaldehyde, acetaldehyde, acrolein and acetone were found during the spring campaign, whereas propionaldehyde and butyraldehyde showed their minimum concentration values during the fall sampling period.

Carbonyls in indoor environments showed a diurnal variation during the spring campaign with higher concentrations in the period comprised from 10:00 to 12:00 h for formaldehyde, acetaldehyde, propionaldehdye and butyraldehyde. Acrolein and acetone showed their maximum concentration values in the periods from 08:00 to10:00 h and from 14:00 to 16:00, respectively. During the summer campaign, all measured carbonyls showed higher concentrations during the period comprised from 10:00 to 12:00 h, except formaldehyde and butyraldehyde, which had their maximum concentrations from 12:00 to 14:00 h. Acetaldehyde showed their maximum indoor concentrations from 10:00 to 12:00 h, the rest of the measured carbonyls had their maximum concentration values in the period comprised from 12:00 to 14:00 h.

### 3.1.3. Diurnal and Seasonal Variation of Outdoor Carbonyls Concentrations

In the present study, measured concentrations of carbonyls in outdoor environments can be related to emissions from vehicular traffic and the increase in the use of oxygenated additives in MZMV. As it can be observed in Table 6, acetone was the dominant carbonyl

in all the sampling sites during the three sampling campaigns. It was expected since acetone is the main end product of oxidation reactions of propanone, isobutane, isopentane and isoalkanes by OH radicals [35,56,57]. Other possible extramural sources associated with acetone are the use of certain products such as enamels, pressed wood sheets (chipboard), paint removers and cigarette smoking. Additionally, car exhaust emissions are another source, so avenues and streets with high traffic congestion detect acetone in the environment [46].

**Table 6.** Diurnal variation of carbonyl outdoor concentrations at the three sampling sites (University-1 UAM, University-2 UDLA and University-3 UNAM) during the three sampling campaigns.

| University-1 (UAM) | | | | | | | | | | | | | | | | | |
|---|---|---|---|---|---|---|---|---|---|---|---|---|---|---|---|---|---|
| **Season:** | | | **Spring** | | | | | | **Summer** | | | | | | **Fall** | | |
| **Sampling Period** | Outdoor Carbonyls concentrations ($\mu$g/m$^3$) | | | | | | | | | | | | | | | | |
| | **FA** | **AA** | **ACR** | **AC** | **PR** | **BUT** | **FA** | **AA** | **ACR** | **AC** | **PR** | **BUT** | **FA** | **AA** | **ACR** | **AC** | **PR** | **BUT** |
| 8–10 | 0.96 | 1.26 | 0.96 | 3.47 | 0.37 | 0.84 | 2.19 | 1.90 | 2.50 | 14.88 | 0.47 | 1.89 | 1.94 | 2.31 | 1.59 | 6.88 | 0.80 | 1.61 |
| 10–12 | 0.93 | 1.14 | 0.68 | 2.46 | 0.31 | 0.73 | 1.58 | 1.70 | 1.93 | 5.30 | 0.47 | 1.16 | 1.66 | 1.79 | 0.06 | 5.11 | 0.65 | 1.09 |
| 12–14 | 0.91 | 1.16 | 0.05 | 2.62 | 0.30 | 0.90 | 1.23 | 1.61 | 2.22 | 11.80 | 0.39 | 0.96 | 1.83 | 1.14 | 0.23 | 3.72 | 0.30 | 0.74 |
| 14–16 | 0.70 | 0.90 | 0.09 | 2.42 | 0.23 | 0.69 | * | * | * | * | * | * | * | * | * | * | * | * |
| **University-2 (UDLA)** | | | | | | | | | | | | | | | | | | |
| **Season:** | | | **Spring** | | | | | | **Summer** | | | | | | **Fall** | | |
| Sampling period | Outdoor Carbonyls concentrations ($\mu$g/m$^3$) | | | | | | | | | | | | | | | | |
| | **FA** | **AA** | **ACR** | **AC** | **PR** | **BUT** | **FA** | **AA** | **ACR** | **AC** | **PR** | **BUT** | **FA** | **AA** | **ACR** | **AC** | **PR** | **BUT** |
| 8–10 | 0.84 | 1.00 | 0.23 | 2.88 | 0.23 | 0.49 | 1.25 | 54.74 | 11.03 | 7.51 | 0.02 | 1.78 | 2.08 | 2.77 | 0.63 | 6.21 | 1.10 | - |
| 10–12 | 1.41 | 1.76 | 0.06 | 9.00 | 0.42 | 1.09 | 1.66 | 3.18 | 9.17 | 9.62 | 0.53 | 2.14 | 2.98 | 4.17 | 0.68 | 11.23 | 1.29 | - |
| 12–14 | 1.42 | 1.55 | - | 5.78 | 0.34 | 1.16 | 2.02 | 2.98 | 8.11 | 9.30 | 0.60 | 2.37 | 1.15 | 1.66 | 0.63 | 5.03 | 0.38 | - |
| 14–16 | 1.39 | 1.45 | - | 5.56 | 0.27 | 1.12 | * | * | * | * | * | * | * | * | * | * | * | * |
| **University-3 (UNAM)** | | | | | | | | | | | | | | | | | | |
| **Season:** | | | **Spring** | | | | | | **Summer** | | | | | | **Fall** | | |
| **Sampling period** | Outdoor Carbonyls concentrations ($\mu$g/m$^3$) | | | | | | | | | | | | | | | | |
| | **FA** | **AA** | **ACR** | **AC** | **PR** | **BUT** | **FA** | **AA** | **ACR** | **AC** | **PR** | **BUT** | **FA** | **AA** | **ACR** | **AC** | **PR** | **BUT** |
| 8–10 | 0.76 | 1.41 | 1.02 | 4.11 | 0.39 | 0.97 | 0.70 | 1.72 | 2.67 | 2.27 | 0.19 | 0.81 | 3.55 | 10.04 | - | 33.72 | 1.94 | 2.59 |
| 10–12 | 0.89 | 1.81 | 0.51 | 4.39 | 0.52 | 1.31 | 0.64 | 1.33 | 1.14 | 2.46 | 0.70 | 1.03 | 3.19 | 10.77 | - | 13.94 | 2.11 | 2.92 |
| 12–14 | 0.67 | 1.48 | 0.06 | 3.90 | 0.34 | 1.11 | 0.64 | 1.01 | 1.65 | 2.05 | 0.20 | 0.83 | 3.16 | 9.45 | - | 1.59 | 1.78 | 2.71 |
| 14–16 | 0.45 | 1.20 | 0.51 | 3.52 | 0.25 | 0.98 | * | * | * | * | * | * | * | * | * | * | * | * |

FA: Formaldehyde; AA: Acetaldehyde; ACR: Acrolein; AC: Acetone; PR: Propionaldehyde; BUT: Butyraldehyde; -: Lower than the limit of detection: *: Not measured.

High concentrations of acetaldehyde were observed, especially during summer. Eventually, the high levels found of this contaminant are related to the increased use of mixtures of gasoline and ethanol fuels in MZMV [9,58]. Seasonal variation in carbonyl outdoor concentrations is related to meteorological conditions such as wind speed and solar radiation. The first is explained by the dispersion and transport of pollutants, while the second promotes the formation of secondary organic compounds and oxidizing radicals that cause the formation of extramural carbonyls. In this evaluation it was observed that carbonyl concentrations are compromised with seasonal meteorological changes.

University-1 (UAM)

In this site, during the spring campaign, all measured carbonyls had a clear diurnal pattern, with the highest outdoor concentrations in the period from 08:00 to 10:00 h, excepting butyraldehyde in the period comprised between 12:00 h and 14:00 h. During the summer and the fall sampling campaigns, all measured carbonyls showed their maximum concentration values between 08:00 h and 10:00 h. There was a clear seasonal variation in outdoor carbonyl concentrations: formaldehyde, acetaldehyde, acetone and butyraldehyde showed their maximum concentration values in indoor environments during the summer

campaign, whereas acrolein and propionaldehyde showed higher concentrations during the summer period.

University-2 (UDLA)

Outdoor carbonyl concentrations in this site had a clear seasonal pattern (Table 6), with higher concentrations during the summer campaign for acetaldehyde, acrolein and butyraldehyde, and higher levels of concentration for formaldehyde and propionaldehyde during the fall campaign. Acetone concentrations in outdoor environments remained constant, which means that this carbonyl was homogeneously distributed along the day in this site. During the spring and the summer sampling campaigns (Table 6), formaldehyde, acetone and butyraldehyde showed their maximum concentration values from 12:00 h to 14:00 h. Acetaldehyde and propionaldehyde had higher outdoor concentrations in the period comprised from 10:00 h to 12:00 h during the spring campaign. Acrolein outdoor concentrations during the spring season had a different diurnal behavior in this site, with maximum values from 08:00 h to 10:00 h. During the summer campaign (Table 6), acetaldehyde and acrolein had their maximum concentrations during the period comprised between 08:00 h to 10:00 h, whereas acetone showed its maximum levels of concentrations from 10:00 to 12:00 h. During the fall campaign (Table 6), all measured carbonyls showed higher outdoor concentrations from 10:00 h to 12:00 h, except acetaldehyde, which showed its maximum concentration values from 08:00 h to 10:00 h.

University-3 (UNAM)

In this site, outdoor carbonyl concentrations were higher during the fall sampling campaign, except acrolein, that had their maximum concentrations during the summer period. The lowest value of carbonyls in outdoor environments in this site were registered during the spring campaign. During the spring period (Table 6), all measured carbonyls showed their maximum concentrations from 10:00 h to 12:00 h, except acrolein, which had higher concentration values in the period comprised between 10:00 h and 12:00 h. During the summer campaign (Table 6), formaldehyde, acetaldehyde and acrolein showed higher outdoor concentrations from 08:00 to 10:00 h, whereas acetone, propionaldehyde and butyraldehyde had their maximum concentration values from 10:00 to 12:00 h. During the fall period (Table 6), formaldehyde and acetone were higher during the period comprised between 08:00 and 10:00 h, whereas acetaldehyde, propionaldehyde and butyraldehyde showed their maximum outdoor concentrations from 10:00 to 12:00 h.

*3.2. Meteorology*

In spring, the wind speed was high with a direction predominantly from the north in the University-2 site (UDLA) and from the north-west in the University-1 site (UAM) towards the south of MZMV, which could have produced a dispersion of air pollutants. On the other hand, it is no coincidence that some of the maximum concentrations of carbonyls were reached in the early hours of the morning, because about 60% of UV radiation is received between 10:00 and 14:00 h. Figures 5–7 show the wind rose for the three sampling sites during spring, summer and fall, respectively. At University-1 (UAM), the wind direction was mostly from the northwest with a prevailing wind speed of 3.6 to 2.1 ms$^{-1}$ (Figure 5), while at University-2 (UDLA) the wind direction was mainly from the northwest with a prevailing wind speed of 8.8 to 5.7 ms$^{-1}$ (Figure 5). For the University-3 site (UNAM), it can be observed that the wind has a northerly direction with a prevailing wind speed of 5.7 to 3.6 ms$^{-1}$ (Figure 5). In summer (Figure 6), at the University-3 (UNAM) site, it was observed that the wind direction came mainly from the northwest with a predominant wind speed of 3.6 to 2.1 ms$^{-1.}$ For the University-1 (UAM) site, the wind presented a prevailing wind speed of 3.6 to 2.1 ms$^{-1}$ with a north and northwest direction; with respect to the University-2 (UDLA) site, the wind presented a prevailing wind speed of 5.7 to 3.6 ms$^{-1}$ and a component from the northwest (Figure 6). In fall (Figure 7), at the University-3 (UNAM) site, it was observed that the prevailing wind speed ranged from 3.6 to 2.1 ms$^{-1}$;

for the University-1 (UAM) site, the wind direction came mainly from the north, but in southern occasions with a prevailing wind speed of 5.7 to 3.6 ms$^{-1}$. For the University-2 (UDLA) site, the direction and velocity came predominantly from the northeast and 8.8 to 5.7 ms$^{-1}$, respectively (Figure 7).

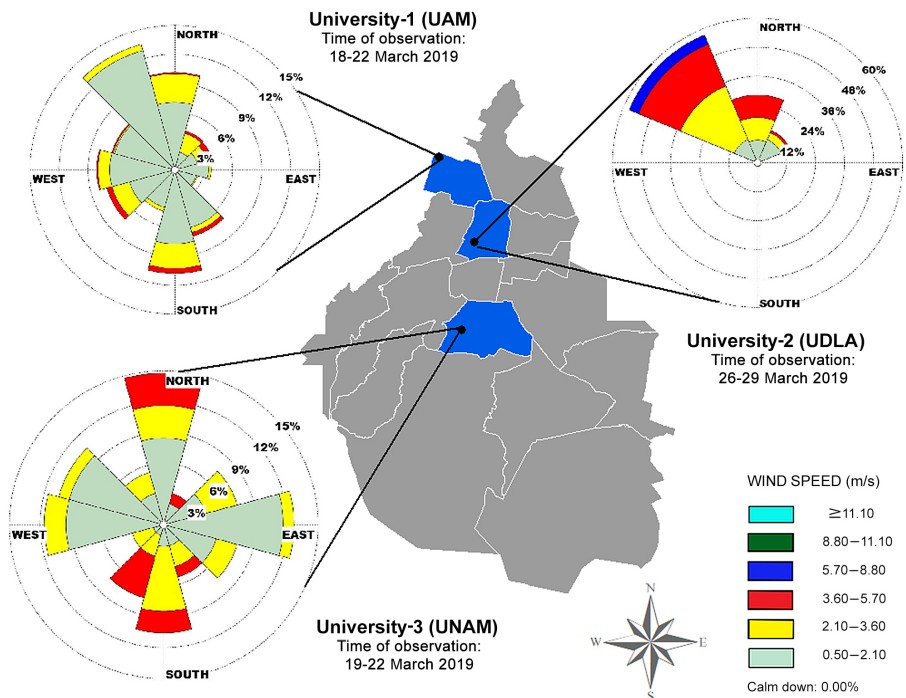

**Figure 5.** Spring sampling campaign wind roses: University-1 (UAM), University-2 (UDLA) and University-3 (UNAM).

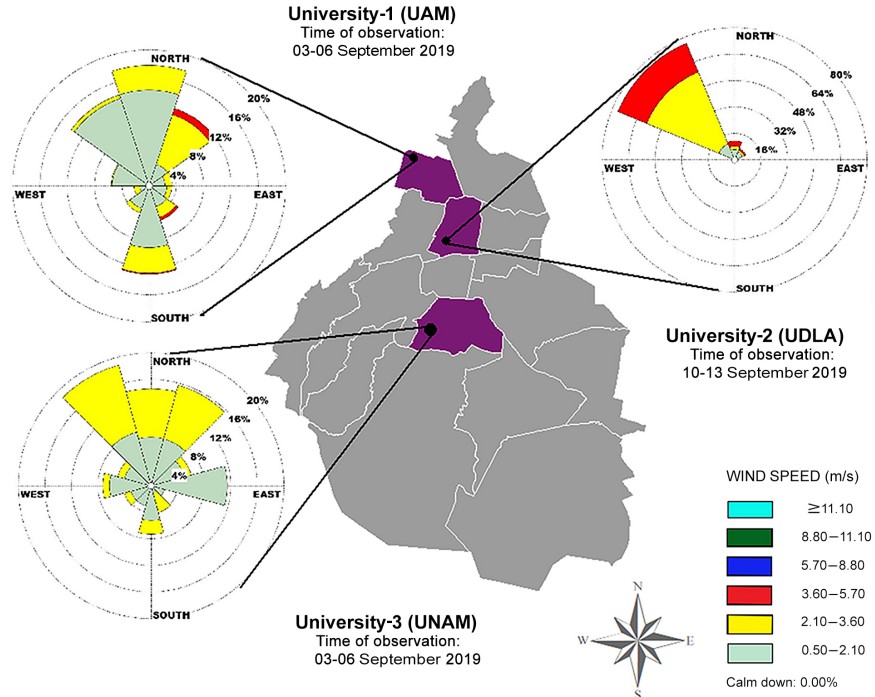

**Figure 6.** Wind roses summer sampling campaign: University-1 (UAM), University-2 (UDLA) and University-3 (UNAM).

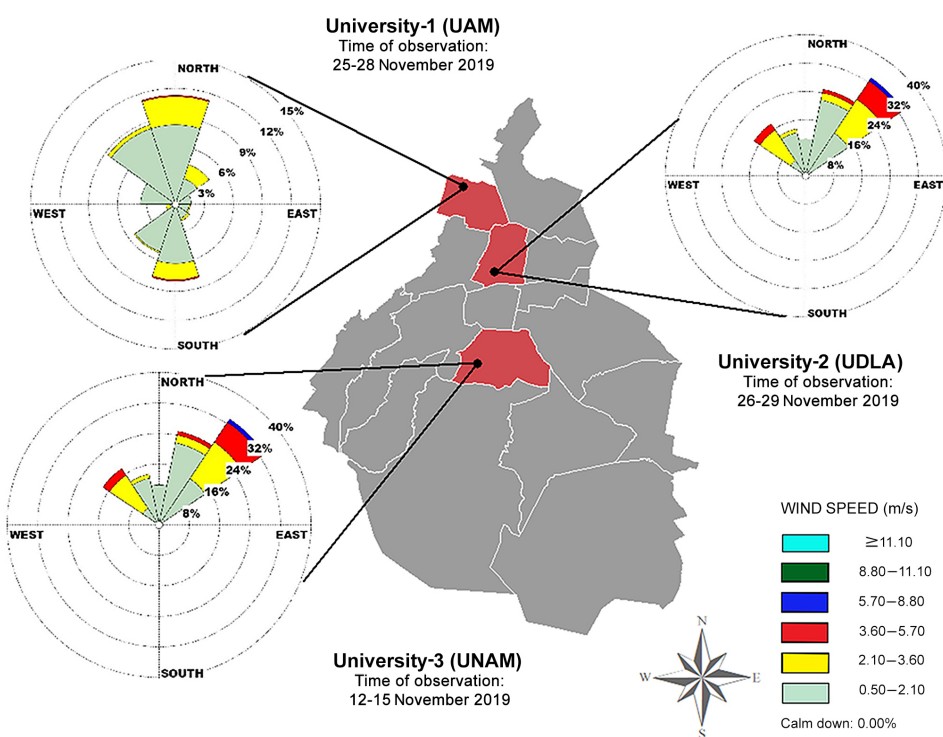

**Figure 7.** Fall sampling campaign wind roses: University-1 (UAM), University-2 (UDLA) and University-3 (UNAM).

### 3.3. Estimated Concentration Ratios (I/E): Concentrations and I/E of Carbonyls

The indoor/outdoor (I/E) concentration ratios of the carbonyl compounds were estimated from the individual values of the concentration for each carbonyl. The average values of the radii for each carbonyl compound evaluated in each campaign appear in Table 7. The average values of the I/E ratio were greater than 1 (I/E > 1) during the summer and the spring and greater than those obtained during the fall. High temperatures and humidity probably influenced emissions from materials and furniture found at sampling sites and reactions indoors that accelerate ozone oxidation. In 90% of the sites, the concentrations presented an I/E ratio > 1 except for acrolein during the summer (I/E = 0.62) and acetaldehyde during the fall with a value of (I/E = 0.55). When I/E ratios were analyzed, it was found that the maximum ratio of 21.8 corresponds to formaldehyde during the spring. The average values of the concentration ratios (I/E) for each carbonyl compound for each site are shown in Table 8, according with this, a variation in the extramural/intramural ratio levels was found.

**Table 7.** The indoor/outdoor (I/E) concentration ratios of the carbonyl compounds.

| Concentration (µg m$^{-3}$) | Spring | Summer | Fall |
|---|---|---|---|
| Acetone | 2.81 | 4.21 | 1.76 |
| Formaldehyde | 21.8 | 5.63 | 3.08 |
| Acetaldehyde | 1.51 | 1.22 | 0.55 |
| Acrolein | 1.19 | 0.62 | 13.25 |
| Propionaldehyde | 1.3 | 1.8 | 0.71 |
| Butyraldehyde | 5.19 | 1.31 | 0.19 |

### 3.4. Health Risk Assessment

The estimated cancer risk coefficients values (LTCR) and daily exposure values for formaldehyde and acetaldehyde (in indoor and outdoor environments) are shown in Table 9. All the sampling sites showed values for LTCR higher than those recommended

values for formaldehyde and acetaldehyde. It can be observed that formaldehyde in indoor environments exceeded the recommended value for LTCR by EPA ($1 \times 10^{-6}$) and WHO ($1 \times 10^{-5}$) in University-1 (UAM) and University-2 (UDLA). LTCR average values for acetaldehyde in indoor environments exceed the maximum permissible levels established by EPA and WHO, associated with significant cancer risk [59,60], in University-1 (UAM) and University-2 (UDLA). For outdoor environments, formaldehyde presented the highest average value for LTCR in University-1 (UAM), exceeding the recommended limits. University-2 (UDLA) and University-3 (UNAM) presented average values for LTCR that exceeded the permissible levels established by WHO for formaldehyde. Acetaldehyde in outdoor environments for the three sampling sites did not exceed the established value by EPA, but it presented higher values of LTCR that the recommended limit established by WHO. Therefore, considering these results, there is a possible risk to developing cancer in lifetime, associated to the exposure to formaldehyde and acetaldehyde by inhalation in the three sampling sites.

**Table 8.** The indoor/outdoor (I/E) concentration ratios of the carbonyl compounds for each sampling site.

| Concentration ($\mu g\ m^{-3}$) | University-1 (UAM) | University-2 (UDLA) | University-3 (UNAM) |
|---|---|---|---|
| Acetone | 2.01 | 7.51 | 15.59 |
| Formaldehyde | 3.7 | 0.65 | 3.75 |
| Acetaldehyde | 2.6 | 0.51 | 0.06 |
| Acrolein | 4.83 | 0.02 | 1.88 |
| Propionaldehyde | 2.69 | 0.20 | 0.45 |
| Butyraldehyde | 1.55 | 1.35 | 0.32 |

**Table 9.** Average values for cancer risk coefficients in lifetime (LTCR) and average daily exposure values for formaldehyde and acetaldehyde in the three sampling sites.

| Cancer Risk Assessment Parameters | University-1 (UAM) | University-2 (UDLA) | University-3 (UNAM) |
|---|---|---|---|
| E (mg/kg/day) Fomaldehyde Indoor | 0.0016 | 0.0003 | 0.0017 |
| E (mg/kg/day) Acetaldehyde Indoor | 0.0012 | 0.0014 | 0.0006 |
| LTCR Formaldehyde Indoor | $3.2719 \times 10^{-5}$ | $8.193 \times 10^{-6}$ | $3.62 \times 10^{-5}$ |
| LTCR Acetaldehyde Indoor | $1.1735 \times 10^{-5}$ | $1.418 \times 10^{-5}$ | $6.908 \times 10^{-6}$ |
| E (mg/kg/day) Fomaldehyde Outdoor | 0.0178 | 0.0003 | 0.0003 |
| E (mg/kg/day) Acetaldehyde Outdoor | 0.0003 | 0.0009 | 0.0008 |
| LTCR Formaldehyde Outdoor | $3 \times 10^{-4}$ | $7.188 \times 10^{-6}$ | $6.347 \times 10^{-6}$ |
| LTCR Acetaldehyde Outdoor | $2.9763 \times 10^{-6}$ | $9.717 \times 10^{-6}$ | $8.2 \times 10^{-6}$ |

Average non-cancer risks are shown in Table 10 for the measured carbonyls in the three sampling sites, considering both indoor and outdoor environments. As can be observed, acetaldehyde and propionaldehyde did not exceed the maximum permissible limit established by EPA for average HQ values (1.0) in both types of environments (indoor and outdoor). Formaldehyde presented HQ values > 1 in University-3 (UNAM), in outdoor environments. HQ average values for acrolein were higher than 1.0 in outdoor and

indoor environments for the three sampling sites. These results indicate that long-term exposure to acrolein and to formaldehyde may result in adverse health effects and that the exposed population may be in risk of developing non-cancer diseases (cardiovascular and respiratory effects) due to the inhalation of these air pollutants.

**Table 10.** Average non-cancer risk coefficients estimated as Hazard Quotients (HQ) for the measured carbonyls in the three sampling sites.

| Non-Cancer Risk Assessment Parameters | University-1 (UAM) | University-2 (UDLA) | University-3 (UNAM) |
|---|---|---|---|
| HQ Formaldehyde Indoor | $4.95 \times 10^{-1}$ | $1.295 \times 10^{-1}$ | $5.67 \times 10^{1}$ |
| HQ Acetaldehyde Indoor | $4.11 \times 10^{-1}$ | $5.141 \times 10^{-1}$ | $2.5 \times 10^{-1}$ |
| HQ Acrolein Indoor | $2.47 \times 10^{2}$ | $5.7 \times 10^{1}$ | $1.99 \times 10^{1}$ |
| HQ Propionaldehyde Indoor | $1.38 \times 10^{-1}$ | $3.78 \times 10^{-2}$ | $6.96 \times 10^{-1}$ |
| HQ Formaldehyde Oudoor | $1.37 \times 10^{-1}$ | $1.704 \times 10^{-1}$ | $1.504 \times 10^{-1}$ |
| HQ Acetaldehyde Outdoor | $1.62 \times 10^{-1}$ | $5.28 \times 10^{-1}$ | $4.459 \times 10^{-1}$ |
| HQ Acrolein Outdoor | $7.54 \times 10^{1}$ | $1.125 \times 10^{1}$ | $1.05 \times 10^{2}$ |
| HQ Propionaldehyde Outdoor | $5.23 \times 10^{-2}$ | $5.63 \times 10^{-2}$ | $1.03 \times 10^{-1}$ |

## 4. Conclusions

Acetone proved to be the most abundant carbonyl compound present in indoor as well as in outdoor, followed by acetaldehyde. Both carbonyls are related to vehicle exhaust emissions and high photochemical activity. Acetaldehyde is especially associated with the use of oxygenated additives in fuels such as ethanol, methanol or MTBE. In this work, it was found that the southern area of the city had higher values of carbonyls compared to the central and northern areas, which could be due to a dispersion effect of the gases coming from the north zone (high vehicular traffic and industrial zone). The highest concentrations of outdoor carbonyls were observed in the fall season, these concentrations were associated with the high solar radiation during this season, which lead to photochemical formation of carbonyl compounds. Meanwhile indoors, the high concentrations during the summer were associated with high temperatures and a high percentage of relative humidity, conditions that occurred both in the office and in the home. The diurnal variation showed an important participation of the photochemical activity in the formation of carbonyl compounds. Maximum concentrations were observed after 12 h in indoor environments, with a period in which human activities and temperatures increased indoors. The higher carbonyl concentrations found in indoor ambient can be attributed to the intensive use of sanitizers and antiviral liquids during the pandemic period. Regarding this, the health risk assessment results showed that the exposure to formaldehyde and acetaldehyde exceeded the recommended values for the lifetime cancer risk coefficients and that HQ values (non-cancer risk) exceeded the unity for acrolein and formaldehyde in the studied sites, indicating that these pollutants may be a significant health risk for the exposed population. Indoor carbonyl concentrations were observed to be below the reference limits for chronic exposure reported by the OEHHA guidelines in California, CA, USA [60]; except for formaldehyde and acrolein, which is related to symptoms such as nose and throat irritation, and eye irritation, among other discomforts. It suggests that the exposure to these compounds in indoor ambient could exacerbate the symptoms associated with COVID-19. It is important to improve the control measures in the studied zone in order to reduce the carbonyl emissions both in indoor and outdoor environments with the aim of protecting the health of the population.

**Author Contributions:** Conceptualization, R.G. and V.M.A.; methodology, R.G. and S.S.G.; software, R.G.; validation, G.A.; formal analysis, S.S.G. and G.A.; investigation, R.G., S.S.G., G.A., R.T.J., A.G.R., J.G.C., R.M.C. and V.M.A.; writing—original draft preparation, R.G. and V.M.A.; writing—review and editing, R.G.; visualization, R.G.; supervision, R.G. and V.M.A.; project administration, R.G. All authors have read and agreed to the published version of the manuscript.

**Funding:** This work was funded by grant PAPIIT-UNAM IN104519.

**Acknowledgments:** We thank Moises López, Manuel García, Miguel Flores-Espinosa, for their valuable reviewing and suggestions on the manuscript.

**Conflicts of Interest:** The authors declare no conflict of interest.

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
