# Peer review of "Measurement of Indoor-Outdoor Carbonyls in Three Different Universities Located in the Metropolitan Zone of Mexico Valley during the First Period of Confinements Due to COVID-19"

_atmosphere, doi:10.3390/atmos13101560_

Round 1

Reviewer 1 Report

The manuscript proposed by Garcìa et al is very interesting and useful, exploring the indoor and outdoor leveles of carbonyls concentrations during the first period of COVID-19 confinements, consindering also the cancer risk coefficient and the non-cancer risk coefficients.

For me the manuscript is well written and structured, with short and coincise but clear introduction and materials and methods sections.

So I just have some minor comments and advises:

i) lines 109-111. I think would be easier to understand for the readers, if the authors can explain further the collection efficiency, mabye adding some references.

ii) lines 118-120. I suggest to add some more details about the calibration. What kind of mixture? Producer, ppb range, etc.

iii) lines 192-196. This is was a little confusing for me, maybe it is a typo in the writing:

1) 192-194, the authors state that there were differencies in the concentrations of all carbonyls among the site execpt acrolein and that this confirmed the different nature of the its source among the sites. The logic does not help me here. If all the other compounds concentrations change between the three sites and that of acrolein not, maybe it is true the contrary? That the acrolein source is always the same?

2) 194-196, this statament it is confusing to me, it seems incomplete. What it does mean?

IV) All the results part it was a bit tedious, repetitive and long for me. I suggest to further summarize it and then try to convey all the informations and details about the differences in the carbonyls concentrations among the hours of the day, the sites and seasons, in one or more summary tables. I think this can shorten the results part and make it more fluent and readable.

Author Response

Please, see the attached file

Reviewer 2 Report

Measurement and analysis of Indoor-outdoor carbonyls in three different locations of Mexico during the first period of Covid-19 has been presented. The subject is interesting and important. So, the manuscript is potentially publishable. There are some points which should be considered and discussed in the revised version for further completion of the work, as mentioned below:

1.       Carbonyls can induce hypoxic-ischemic encephalopathy which can also be exacerbated by Covid-19 (see, for example, [ACS Applied Nano Materials 4 (2021) 11386-11412]). This issue should be addressed in the revised version.

2.       Could the authors comment on the probable effects of carbonyls on the activity of SARS-Cov-2 virus? For example, carbonyls show good attachment to viruses (see, e.g., [Anal. Chem. 2018, 90, 12, 7139–7147]). Moreover, carbonyl groups present on graphitized oxide sheets can act as effective antiviral agents (see, for example, [Catalysts 12 (2022) 667]). This point should be addressed and discussed in the revised version.  

3.       Air humidity can affect the (biological/chemical) activity of carbonyls. So, could the comments on the air humidity of indoor and outdoor of the locations and also its possible effects on human health?

Author Response

Please, see the attached file

Round 2

Reviewer 2 Report

Revisions are acceptable.